# Prostate-Specific Membrane Antigen Radioligand Therapy in Non-Prostate Cancers: Where Do We Stand?

**DOI:** 10.3390/bioengineering11070714

**Published:** 2024-07-14

**Authors:** Francesco Dondi, Alberto Miceli, Guido Rovera, Vanessa Feudo, Claudia Battisti, Maria Rondini, Andrea Marongiu, Antonio Mura, Riccardo Camedda, Maria Silvia De Feo, Miriam Conte, Joana Gorica, Cristina Ferrari, Anna Giulia Nappi, Giulia Santo

**Affiliations:** 1Nuclear Medicine Unit, ASST Spedali Civili di Brescia, 25123 Brescia, Italy; francesco.dondi@unibs.it; 2Nuclear Medicine Unit, Azienda Ospedaliero-Universitaria SS. Antonio e Biagio e Cesare Arrigo, 15121 Alessandria, Italy; alberto.miceli@ospedale.al.it; 3Nuclear Medicine Division, Department of Medical Sciences, University of Turin, 10126 Turin, Italy; guido.rovera@unito.it; 4Unit of Nuclear Medicine, Aosta Regional Hospital, 11100 Aosta, Italy; vfeudo@ausl.vda.it; 5Section of Nuclear Medicine, Interdisciplinary Department of Medicine, University of Bari “Aldo Moro”, Piazza Giulio Cesare 11, 70124 Bari, Italy; claudiabattisti93@gmail.com (C.B.); ferrari_cristina@inwind.it (C.F.); 6Nuclear Medicine Unit, Fondazione Istituto G. Giglio, 90015 Cefalù, Italy; maria.rondini01@ateneopv.it; 7Unit of Nuclear Medicine, Department of Medical, Surgical and Experimental Sciences, University of Sassari, 07100 Sassari, Italy; amarongiu2@uniss.it (A.M.); a.mura203@studenti.uniss.it (A.M.); 8Nuclear Medicine Unit, Department of Biomedicine and Prevention, University of Rome “Tor Vergata”, 00133 Rome, Italy; riccardo.camedda@gmail.com; 9Nuclear Medicine Unit, Department of Radiological Sciences, Oncology and Anatomo-Pathology, Sapienza University of Rome, 00185 Rome, Italy; mariasilvia.defeo@uniroma1.it (M.S.D.F.); miriam.conte@uniroma1.it (M.C.); joana.gorica@uniroma1.it (J.G.); 10Nuclear Medicine Unit, Department of Experimental and Clinical Medicine, “Magna Graecia” University of Catanzaro, 88100 Catanzaro, Italy; giulia.santo@unicz.it

**Keywords:** PSMA, radioligand therapy, solid tumours, non-prostate, luthetium-177

## Abstract

Introduction: The term theragnostic refers to the combination of a predictive imaging biomarker with a therapeutic agent. The promising application of prostate-specific membrane antigen (PSMA)-based radiopharmaceuticals in the imaging and treatment of prostate cancer (PCa) patients opens the way to investigate a possible role of PSMA-based radiopharmaceuticals in cancers beyond the prostate. Therefore, the aim of this review was to evaluate the role of ^177^Lu-PSMA radioligand therapy (RLT) in malignancies other than prostate cancer by evaluating preclinical, clinical studies, and ongoing clinical trials. Methods: An extensive literature search was performed in three different databases using different combinations of the following terms: “Lu-PSMA”, “^177^Lu-PSMA”, “preclinical”, “mouse”, “salivary gland cancer”, “breast cancer”, “glioblastoma”, “solid tumour”, “renal cell carcinoma”, “HCC”, “thyroid”, “salivary”, “radioligand therapy”, and “lutetium-177”. The search had no beginning date limit and was updated to April 2024. Only articles written in English were included in this review. Results: A total of four preclinical studies were selected (breast cancer model *n* = 3/4). PSMA-RLT significantly reduced cell viability and had anti-angiogenic effects, especially under hypoxic conditions, which increase PSMA binding and uptake. Considering the clinical studies (n = 8), the complexity of evaluating PSMA-RLT in cancers other than prostate cancer was clearly revealed, since in most of the presented cases a sufficient tumour radiation dose was not achieved. However, encouraging results can be found in some types of diseases, such as thyroid cancer. Some clinical trials are still ongoing, and results from prospective larger cohorts of patients are awaited. Conclusions: The need for larger patient cohorts and more RLT cycles administered underscores the need for further comprehensive studies. Given the very preliminary results of both preclinical and clinical studies, ongoing clinical trials in the near future may provide stronger evidence of both the safety and therapeutic efficacy of PSMA-RLT in malignancies other than prostate cancer.

## 1. Introduction

The term “theragnostic” is by definition a combination of the terms “therapy” and “diagnostic” and describes the use of a predictive imaging biomarker in combination with a therapeutic agent [1,2]. From its historic introduction with radioiodine treatment in the 1950s [3], the translation of this concept into various molecular imaging and therapeutic applications has steadily increased over time. In recent years, the success of prostate-specific membrane antigen (PSMA)-based radiopharmaceuticals for the imaging and treatment of prostate cancer (PCa) patients has further enhanced this concept (Figure 1) thanks to the promising opportunities it offers [2]. 

PSMA, also known as glutamate carboxypeptidase II (GCPII), is a transmembrane glycosylated protein with an extracellular domain readily accessible for ligand binding [4]. As its name suggests, it is expressed on the membrane of normal prostate epithelial cells and is overexpressed by most prostate cancer cells (~85%) [5,6], making it a favourable target for both imaging and therapeutic procedures. PSMA has been labelled with various diagnostic radionuclides, such as gallium-68 (^68^Ga) or fluoride-18 (^18^F), and is used for positron emission tomography (PET) imaging in several clinical settings [7,8]. Indeed, several PSMA-based radiotracers such as [^68^Ga]Ga-PSMA-11, [^18^F]-PSMA-1007, and [^18^F]-DCFPyl are now available and have demonstrated their superiority over other radiopharmaceuticals previously used for PCa. Although insufficient evidence has been reported to favour the use of one ligand over another [9], [^68^Ga]Ga-PSMA-11 has shown potentially superior properties in terms of blood pool persistence and background rate accumulation [10,11]. On the other hand, PSMA ligands labelled with lutetium-177 (^177^Lu) represent promising radiopharmaceuticals for specific radioligand therapy (RLT) of castration-resistant prostate cancer (mCRPC) patients [12,13]. ^177^Lu is an isotope with a physical half-life of 6.7 days and the ability to emit β-particles with a maximum and average soft tissue path length of approximately 1.7 and 0.2 mm, respectively, which are responsible for DNA damage and subsequent cancer cell death [14]. The phase 3 VISION trial showed that [^177^Lu]Lu-PSMA-617 (Pluvicto^TM^) therapy prolonged imaging-based progression-free survival (PFS) and overall survival (OS) in patients with advanced PSMA-positive mCRPC [15], leading to final approval by the United States Food and Drug Administration (FDA) [16].

Despite its name, PSMA is not completely specific for prostate cancer. It is physiologically present in some normal tissues such as the small intestine, proximal renal tubules, and salivary glands [17]. In addition, endothelial cells of tumour-associated neovasculature of various neoplasms (e.g., renal cancer, breast cancer, glioblastoma, hepatocellular carcinoma, and lung cancer) have been shown to overexpress PSMA [18,19]. The role of PSMA in the neovasculature of non-prostate tumours has not been fully elucidated. PSMA knockout animals showed a reduction in vascular growth and haemoglobin content compared with the wild type [20]. Similarly, a PSMA inhibitor (i.e., 2-phosphonomethylpentanedioic acid) has been shown to reduce neovascularization. Examination of several extracellular proteins revealed that PSMA-induced vascular invasion is mediated by laminin. Indeed, previous studies have shown that PSMA carboxypeptidase activity leads to endothelial cell activation through the phosphorylation of focal adhesion integrin kinase and subsequent p21-activated kinase (PAK) activation, the several laminin-derived peptides containing carboxy-terminal glutamate moieties (LQE, IEE, and LNE) are substrates of PSMA [20]. In addition, LQ, a product of the enzymatic cleavage of LQE by PSMA, effectively activates endothelial cells in vitro, increases angiogenesis in vivo, and is dependent on integrin β1 activation [20,21].

Thus, the possibility of targeting PSMA opens the way for further investigation of PSMA-based radiopharmaceuticals for imaging and therapeutic purposes in cancers beyond PCa. Several clinical researchers have evaluated the use of PSMA-based imaging in various solid tumours, showing promising results in different clinical settings and also in comparison with other radiopharmaceuticals [22,23,24,25,26]. 

The therapeutic translation of PSMA-based probes in cancers beyond the prostate is still under debate. Namely, the therapeutic effect in PCa is mainly due to PSMA overexpression in PCa cell membranes, whereas in other cancers, the prevalent expression by the tumour-associated endothelium may limit the efficacy of radioligand treatment. 

Therefore, the aim of this review is to assess the role of [^177^Lu]Lu-PSMA therapy in malignancies other than PCa by evaluating preclinical studies (including in vitro and mouse models), clinical studies, and ongoing clinical trials. 

## 2. Search Strategies

A wide literature search was performed using the PubMed/Medline, Scopus, and Google Scholar databases to find any available original articles on the role of [^177^Lu]Lu-PSMA in malignancies other than PCa. The keywords of the inquiry, both as text and MeSH terms variously combined, were “Lu-PSMA”, “^177^Lu PSMA”, “preclinical”, “mouse”, “salivary gland cancer”, “breast cancer” “glioblastoma” “solid tumour”, “renal cell carcinoma”, “HCC”, “thyroid”, “salivary”, “radioligand therapy” and “luthetium-177”. The search had no beginning date limit and was updated until April 2024. Only articles written in the English language were included in the present review. In addition, conference proceedings, editorials, or reviews were excluded. To identify further eligible articles, the references in the retrieved articles were also screened for additional papers. Two reviewers (F.D. and A.M.) independently screened, retrieved, and selected data from each manuscript. Disagreements between the reviewers were solved through the involvement of a third author. For each study considered in this review, data concerning its basic information were retrieved and used to complete Table 1, Table 2 and Table 3.

## 3. Results

### 3.1. Preclinical Studies 

Studies on the preclinical evaluation of treatment by using [^177^Lu]Lu-PSMA in different tumour types are summarized in Table 1. As shown, the lack of in vitro or in vivo experimental models specifically investigating PSMA expression on the neovasculature and on neoplastic cells has limited the development of imaging and therapeutic approaches using PSMA targeting for other tumour models. 

The available literature of preclinical studies based on breast cancer is reported here. A paper by Heesch et al. [27] delved into the analysis of PSMA and its isoforms expression in a panel of 12 triple-negative breast cancer (TNBC) cell lines, including breast cancer stem cells (BCSCs) and tumour-associated endothelial cells. First, the authors demonstrated that the full-length PSMA transcript was present in all TNBC lines apart from line HCC1937. In addition, the highest expression of the PSMAD18 isoform was observed in MDA-MB-231-, BT-20-, SUM1315MO2-, and FL-PSMA-negative HCC1937 cell lines. Stem cells were positive for four out of six isoforms with the PSMAD18, PSM-D, and PSM-E isoforms being the most represented. Moreover, the researchers examined the expression of PSMA on endothelial cell lines isolated from the human umbilical vein (HUV-EC-C, referred to as HUVEC), revealing that the co-culture with TNBC lines induced membrane and cytosol endothelial PSMA expression. Additionally, PSMA was expressed in endothelial cells of the TNBC and HUVEC tumour spheroid model, particularly in hypoxic conditions. The highest uptake of [^68^Ga]Ga-PSMA-11 in TNBC cells was detected in the BCSC1 and BCSC2 lines, with 4.2% and 2.8% after 4 h, respectively. However, the TNBC cell lines showed lower binding and uptake of [^68^Ga]Ga-PSMA-11. Interestingly, hypoxia increased the [^177^Lu]Lu-PSMA-I&T uptake with relevant results in MCF-10A and MDA-MB-231 lines. A high apoptotic rate was seen in the TNBC cell lines when treated with [^177^Lu]Lu-PSMA-I&T and in HUVEC cells when they were placed in co-culture with TNBC. Hypoxia significantly increased the binding and uptake of [^177^Lu]Lu-PSMA in the MCF-10A (0.3% vs. 3%) and MDA-MB-231 (0.4% vs. 3.4%) lines, whereas the prostate cancer cells (also studied for comparison purpose) showed decreased uptake under hypoxic conditions. The authors also investigated whether [^177^Lu]Lu-PSMA-I&T had an apoptotic effect on the TNBC cells and the associated endothelial cells, revealing that cell lines MDA-MB-231 and BT-20 showed a higher apoptotic rate than MCF-10A. [^177^Lu]Lu-PSMA-I&T had no effect on the MCF-10A control. In the HUVEC, the highest apoptosis was detected in co-cultures with TNBC cell lines. The value of PSMA RLT in TNBC was also previously studied by Morgenroth et al. [28], who revealed that MDA-MB231 lines had a remarkable pro-angiogenic ability to induce HUVEC cells to form tubes and induce PSMA expression on endothelial cells. When administered, ^177^Lu-labelled PSMA-617 significantly reduced HUVEC cell viability and had anti-angiogenic effects. Additionally, HUVEC and TNBC cells demonstrated higher uptake of [^68^Ga]Ga-PSMA-11 under hypoxic conditions, as previously underlined. In vivo, radiolabelled PSMA-ligand accumulated specifically in the TNBC xenograft MDA-MB231, whereas no [^68^Ga]Ga-PSMA-11 uptake was identified in the estrogen-sensitive MCF-7 xenograft. The ex vivo immunofluorescence study revealed the presence of PSMA on MDA-MB231 xenograft-associated endothelial cells and TNBC cells.

The therapeutic efficacy of [^177^Lu]Lu-PSMA-I&T in an orthotopic model of TNBC was recently evaluated by Heesch and colleagues [29]. [^177^Lu]Lu-PSMA-I&T was analysed after the intravenous administration of a single dose (60 MBq) or four fractionated doses every 7 days (4 × 15 MBq). Tumour growth was monitored using [^18^F]FDG PET/CT. The tumour volume 30 days after therapy was significantly smaller for the single dose (*p* < 0.001) and fractionated dose (*p* < 0.001) groups compared with the control group. Tumour growth analysis ended on the day of finalization of the last control animal. Interestingly, only control group animals reached the end point for tumour volume, while the remaining animals were finalized solely because of tumour diameter. Statistical analysis revealed significantly smaller tumour volumes at the time of finalization of the single dose- (*p* = 0.003) and fractionated dose-treated (*p* = 0.001) animals compared with the control group (mean tumour volume = 1511, 1088, and 1044 mm^3^ for the control, single dose, and fractionated dose therapy groups, respectively). The calculated relative tumour growth inhibition rates were 38% and 30% for single-dose and fractionated-dose therapy, respectively. The Kaplan–Meier analyses indicated a median survival of 19 d for the control group, 31 d for the single dose, and 28 d for the fractionated dose group (not significant). The apoptotic effect in isolated organs was visualized using ex vivo TUNEL staining. In the tumour tissue, both therapy groups showed a higher number of apoptotic cells compared with the control group. Importantly, this effect was observed exclusively at the tumour edges. The core region of the tumours was highly apoptotic/necrotic in all groups. Considering healthy tissue, the kidney was the only organ that showed some single apoptotic cells. However, these were not caused by the radiation as they also appeared in the control group. For further therapy evaluation, tumour tissue was stained with HIF1α antibody. In the tumour edge regions of the control animals, the hypoxia marker HIF1α was detected in the nucleus and cytoplasm; on the other hand, in the treated animals, HIF1α was preferentially located in the cell nucleus. The tumour core in all study groups was negative for HIF1α. The binding of [^177^Lu]Lu-PSMA-I&T was verified via Autoradiography (AURA) analysis, Microautoradiography (mAURA), and immunostaining of tumour tissue sections. The control tumour demonstrated higher [^177^Lu]Lu-PSMA-I&T binding compared with the therapy groups. Noteworthily, tumours in the therapy groups showed more necrotic tissue and less viable areas compared with the control. The mAURA with [^177^Lu]Lu-PSMA-I&T demonstrated a clear accumulation of silver grains on the walls of the blood vessels in all groups.

The value of PSMA RLT in hepatocellular carcinoma (HCC) was investigated by Lu et al. [30]. [^177^Lu]Lu-PSMA-617 and [^177^Lu]Lu–Evans blue (EB)–PSMA-617 were evaluated using a PSMA-positive HepG2 human HCC subcutaneous xenograft mouse model with the injection of the two tracers. Single-photon emission computed tomography/computed tomography (SPECT/CT) revealed higher uptake for [^177^Lu]Lu-EB-PSMA-617 compared with [^177^Lu]Lu-PSMA-617. For the specific setting of RLT, four groups were identified based on tracer type and administered dose, including 37 MBq of [^177^Lu]Lu-PSMA-617, 18.5 MBq of [^177^Lu]Lu-PSMA-617, 7.4 MBq of [^177^Lu]Lu-EB-PSMA-617, and a saline as control. [^177^Lu]Lu-PSMA-617 was capable of hindering tumour growth at all the established doses, and no toxicity was observed, therefore demonstrating the safety and efficacy of both radiopharmaceuticals.
bioengineering-11-00714-t001_Table 1Table 1Preclinical studies on PSMA-RLT in cancer types beyond the prostate.First Author [Ref.]YearType of DiseaseCell Lines/Mouse ModelMain FindingsHeesch A [27]2023Breast cancerEndothelial cell line (HUVEC), benign breast epithelial cell line (MCF-10A), PCa cell line (LNCaP), and TNBC cell lines (MDA-MB-231, MDA-MB-468, BT-20, Hs578T, SUM149PT, SUM1315MO2, HCC1937)PSMA expression was detected in 91% of the investigated TNBC cell lines. Hypoxic conditions significantly increased the uptake of [^177^Lu]Lu-PSMA in MDA-MB-231 (0.4% vs. 3.4%) and MCF-10A (0.3% vs. 3.0%). [^177^Lu]Lu-PSMA-induced apoptosis rates were highest in BT-20- and MDA-MB-231-associated endothelial cells.Morgenroth A[28]2019Breast cancer- Human breast cancer cell lines (MDA-MB 231 and MCF-7); endothelial cells (HUVEC).- Subcutaneous xenograft.[^177^Lu]Lu-PSMA-617 impaired the vitality and angiogenic potential of cells. In vivo, PSMA accumulated specifically in triple-negative breast cancer xenografts.Heesch A [29]2024Breast cancer- Human breast cancer cell lines MDA-MB-231.- Orthotopic xenograft.The tumour volume 30 days after therapy was significantly smaller for the single-dose (*p* < 0.001) and fractionated dose (*p* < 0.001) groups compared with the control. In the tumour tissue, both therapy groups showed a higher amount of apoptotic cells compared with the control groupLu Q [30]2023HCC- Human hepatocellular cancer cells HepG2.- Subcutaneous xenograft.Tumour growth was significantly suppressed in the 37 MBq [^177^Lu]Lu-PSMA-617, 18.5 MBq [^177^Lu]Lu-PSMA-617, and 7.4 MBq [^177^Lu]Lu-EB-PSMA-617 groups compared with the saline group. Median survival was 40, 44, 43, and 30 days, respectively. No healthy organ toxicity was observed.Abbreviations: PCa, prostate cancer; TNBC, triple-negative breast cancer; PSMA, prostate-specific membrane antigen; HCC, hepatocellular carcinoma.


### 3.2. Clinical Studies

Recently, the potential role of PSMA RLT in non-prostate cancer settings has also been evaluated in a few preliminary clinical studies, mostly related to small cohorts of patients or single case reports (Table 2). Civan et al. conducted a single-centre retrospective study of twenty-eight patients diagnosed with PSMA-positive salivary gland tumours, demonstrating a higher diagnostic accuracy of PSMA PET/CT compared with CT and the feasibility of RLT (PSMA expression level was higher than liver in six patients—25%). Of these patients, only five subsequently underwent [^177^Lu]Lu-PSMA-617 and dosimetry evaluation. The therapy was well tolerated with no major adverse events. However, PSMA-RLT was discontinued after one cycle in three of five patients because insufficient tumour doses were achieved [31]. Klein Nulent et al. investigated the use of [^177^Lu]Lu-PSMA-617 as a last-resort treatment for patients with recurrent metastatic salivary gland cancers and tumour-related discomfort. RLT eligibility was confirmed when tumour targeting was greater than the liver maximum standardized uptake value (SUV_max_) on [^68^Ga]Ga-PSMA-11 PET/CT. Six patients (four with adenoid cystic carcinoma, one with adenocarcinoma NOS, and one with acinic cell carcinoma) underwent a four-cycle protocol, receiving doses of 6–7.4 GBq of [^177^Lu]Lu-PSMA-617 every 6–8 weeks. In two patients, the radiological response was observed, showing either stable disease or partial response, and four patients reported immediate relief of tumour-related symptoms. Moreover, the study revealed minimal side effects (grade 1–2 fatigue, nausea, bone pain, and xerostomia), indicating that [^177^Lu]-PSMA-617 was well-tolerated [32]. Similar conclusions were also demonstrated in a patient with adenoid cystic carcinoma of the parotid reported by Simsek and colleagues. Namely, treatment was well tolerated with no side effects and significant but not complete pain relief was reported by the patient [33]. Another study by Wang et al. investigated the efficacy of RLT using [^177^Lu]Lu-EB-PSMA-617 in four patients diagnosed with adenoid cystic carcinoma. One patient received a dosage of approximately 1.85 GBq per cycle for up to three cycles, while the other three received a dosage for just one cycle. The study showed promising results, with two patients exhibiting remarkable responses to RLT (one complete response, one partial response). Conversely, the other two patients demonstrated heterogeneous results, with reduced uptake in recurrent tumour, lung, and visceral metastases, while increased uptake in bone metastases [34].

Despite diagnostic–therapeutic advances, high-grade gliomas (HGGs) still have a poor prognosis. Graef et al. conducted a study with ^177^Lu-PSMA RLT with intratherapeutic dosimetry. Although twenty patients showed increased tracer uptake in tumours on [^68^Ga]Ga-PSMA-11 PET/magnetic resonance imaging (MRI), only three patients were eligible for the treatment according to the European Association of Nuclear Medicine (EANM) guideline on selecting eligible patients with mCRPC (tumour-to-liver ratio > 1). In these patients, two cycles of ^177^Lu-PSMA therapy were administered, with a median treatment activity of 6.03 GBq. However, the tumour absorbed dose (median dose of 0.56 Gy) was too low to be effective, especially compared with the median dose achieved in external radiation therapy [35]. Similarly, in the clinical study conducted by Truckenmueller et al., three of twenty patients with different grades of glioma (recurrent and heavily pre-treated), presented a tumour-to-liver ratio > 1 and qualified for ^177^Lu-PSMA RLT as a late-line therapy after recurrence. The treatment consisted of two cycles, with a median time interval of 10 weeks and a standard median dose of 6.03 GBq. Despite no observed treatment-related toxicity, all patients required palliative care because of worsening clinical and neurological conditions after the second cycle. However, efficacy data could not be properly assessed because of the short follow-up period of 15 weeks [36].

Hirmas et al. conducted a retrospective study in forty patients with HCC submitted to [^68^Ga]Ga-PSMA-11 PET/CT imaging. Even though [^68^Ga]Ga-PSMA-11 PET/CT demonstrated higher accuracy than CT in the detection of HCC (above all for metastasis) and was associated with a management change in about half the patient cohort, only two patients presented liver lesions with significant uptake on [^68^Ga]Ga-PSMA-11 PET scans to propose RLT since no other local or systemic treatment options were viable. SPECT/CT-based dosimetry revealed that the tumour received a radiation dose at least ten times lower than what is typically achieved with one cycle of external-beam radiation therapy for HCC. Consequently, this treatment approach did not yield the prospected effectiveness, and RLT was halted after just one cycle for both patients [37].

Numerous studies have highlighted the presence of PSMA in the neovasculature of thyroid malignancies [38,39]. Given the bleak prognosis and scarcity of treatment perspectives for patients with anaplastic (ATC) and poorly differentiated (PDTC) thyroid carcinoma, Wächter et al. explored the theranostic potential of PSMA expression in twenty-two ATC and six PDTC patients. Only a patient diagnosed with PDTC exhibited solely PSMA-positive lesions. This patient underwent two cycles of [^177^Lu]Lu-PSMA RLT, which effectively maintained disease stability for a duration of seven months. [^68^Ga]Ga-PMSA PET/CT was performed to complete [^18^F]FDG PET/CT in identifying missed lesions, aiding the selection of patients with ATC and PDTC suitable for RLT [40]. In the study by de Vries et al., five patients with radioactive iodine-refractory DTC underwent [^68^Ga]Ga-PSMA PET/CT to determine whether they were eligible for therapy with [^177^Lu]Lu-PSMA-617 (visually assessed, no pre-defined SUV cut-off values). Four patients had papillary thyroid carcinoma (PTC), and one patient had the follicular variant of PTC. Three patients were deemed eligible for treatment with [^177^Lu]Lu-PSMA-617, but only two were treated with two cycles of 6.0 GBq [^177^Lu]Lu-PSMA-617. One patient showed disease progression on imaging one month later, while her thyroglobulin (Tg) values gradually increased from 18 to 63 μg/L in the months after treatment. The other patient showed a partial, temporary response of lung and liver metastases. Tg levels initially decreased from 17 to 9 μg/L. However, seven months after treatment, there was disease progression on imaging, and Tg levels increased to 14 μg/L [41].

Some case examples were also published on the use of PSMA-RLT in leiomyosarcomas patients. Namely, Digklia et al. [42] and Jüptner et al. [43] reported on two cases of uterus and vena cava inferior leiomyosarcoma, respectively. Even without any adverse side effects reported (such as all clinical cases reported, i.e., [44]), radionuclide therapy did not play a role in the consequential progression of the disease and the discontinuation of the treatment.
bioengineering-11-00714-t002_Table 2Table 2Clinical studies on PSMA-RLT in cancer types beyond prostate cancer.First Author [Ref.]YearType of DiseasePatients^177^Lu-PSMA RLTMain FindingsCivan C [31]2023Salivary gland tumours5One cycle in three patients and RLT completed in two patients; 6.8 ± 1.4 GBq; time interval 6 weeksPSMA RLT was well tolerated and stabilized disease in one patient. However, frequent discontinuation after one PSMA RLT cycle and low tumor absorbed doses were shown.Klein Nulent TJW [32]2021Metastatic salivary gland tumours6One to four cycles; 6.0–7.4 GBq; interval time 6–8 weeksWhen tumour targeting was sufficient, palliative PSMA RLT of advanced/metastasized salivary gland cancer may cause a significant relief of tumour-associated discomfort and may induce disease control in one-third of the cases.Has Simsek D [33]2019Adenoid cystic carcinoma of the parotid1One cycle, 7.5 GBqThe treatment was well tolerated with no side effects reported. Significant but not complete pain relief was expressed by the patient.Wang G [34]2022Adenoid cystic carcinoma4Up to three cycles; 1.85 GBq; interval time 8–10 weeksPSMA RLT based on [^177^Lu]Lu-EB-PSMA-617 may be a promising treatment for adenoid cystic carcinoma.Graef J [35]2023High-grade glioma3Two cycles; median activity of 6.03 GBq (IQR 5.74–6.10)In high-grade glioma, a minority of patients were eligible for PSMA-RLT, and the tumour dose was too low for a sufficient therapeutic effect.Truckenmueller P [36]2022High-grade glioma3Two cycles; median activity of 6.03 GBq (5.74–6.10); time interval 9–11 weeksOnly a minor proportion of the patients were eligible for PSMA-RLT based on the TBR_max_ threshold.Hirmas N [37]2021Hepatocellular carcinoma2One cycle; 5.9–6.2 GBqPSMA-RLT was not effective since it did not yield a sufficient tumour radiation dose.Wächter S [40]2021Anaplastic and poorly differentiated thyroid carcinoma1Two cycles, 6.3 GBq e 7.4 GBq, time interval 8 weeksPSMA-targeted therapy could be used as an alternative option in selected patients if they showed progression after established therapeutic lines.de Vries LH [41]2020Radioactive iodine-refractory differentiated thyroid cancer2Two cycles; 6 GBq, time interval 6 and 11 weeksPSMA-RLT showed a modest, temporary response.Digklia A [42]2022Uterine leiomyosarcoma1Two cycles (2 months apart) combined with 240 mg of nivolumab (every 2 weeks)At 6 months post-treatment, a reduction in the tumor growth rate (TGR (%/month) from 36.46%/m to 11.25%/m was shown.Jüptner [43]2019Vena cava leiomyosarcoma1One cycle, 6 GBqTreatment was well tolerated. However, because of the week retention of the radiotracer, the therapy was discontinued, and no further treatment cycles were arranged.Simsek [44]2021Testicular mixed germ cell tumour1One cycle, 7.5 GBqTreatment was well tolerated without any adverse effects. However, the disease progressed.Abbreviations: PSMA, prostate-specific membrane antigen; RLT, radioligand therapy; TBR, tumour-to-background ratio.


### 3.3. Ongoing Clinical Trials

To date, seven recorded clinical trials are investigating PSMA-RLT in non-prostatic cancers (Table 3). At the time of our analysis, two trials had no reported progress status (NCT04801264–NCT05170555), one trial completed the recruitment phase (NCT04291300), and the remaining trials are still ongoing, being started in 2023 (NCT05867615–NCT05644080–NCT05420727–NCT06059014). Five out of seven trials are conducted by authors from Europe (NCT04291300–NCT05867615–NCT05644080–NCT05420727–NCT06059014) and two studies are performed by researchers from China (NCT05170555–NCT04801264). The two latest trials, having primary diagnostic purposes, are based on the therapeutic administration of ^177^Lu-EB-PSMA-617 at the lowest activity per cycle among recorded clinical trials, while all the European studies have primary therapeutic outcomes and involve the administration of ^177^Lu-PSMA-I&T or its generic formulations (^177^Lu-ITG-PSMA-1 and ^177^Lu-PSMA-1) as therapeutic agents. The number of estimated participants ranges from 10 to 100, the latter corresponding to the Phase 2 study conducted by the “Istituto Scientifico Romagnolo per lo Studio e la Cura dei Tumori”, enrolling subjects with different types of advanced/metastatic solid tumours, including patients with bone metastatic prostate cancer (NCT05867615-LUBASKET). This Italian study, with an estimated completion date in December 2028, represents the sole basket trial, since all other studies are focused on specific tumours. The assessment of the role of PSMA-RLT in advanced salivary gland cancer has resulted in two clinical trials, the Early Phase 1 trial by “Peking Union Medical College Hospital” selected only adenoid cystic carcinoma (ACC) (NCT04801264), which led to the publication of the above-mentioned pilot study in 2022 [33], and the “Radboud University Medical Center” Phase 2 study completed in February 2023 with two different cohorts, including subjects with ACC and patients with salivary duct carcinoma (SDC) (NCT04291300–LUPSA). Renal cancer is also widely represented, being involved in two out of seven trials, specifically the study by “Peking Union Medical College Hospital”, which is evaluating patients with advanced renal cell carcinoma (RCC) (NCT05170555), and the multicentric Phase 1/Phase 2 study by the “Centre Leon Berard”, which is enrolling subjects with metastatic clear cell RCC (mccRCC) and has April 2027 as its estimated completion date (NCT06059014–PRadR). The feasibility of a vascular disruption approach through PSMA-RLT for patients with soft tissue sarcoma is investigated in a Phase 1 clinical trial by the “Centre Hospitalier Universitaire Vaudois” (NCT05420727—ThernSarc). Similarly, in HGG, the pilot study by “St. Olavs Hospital” in collaboration with “National Taiwan Normal University” aims to evaluate PSMA-based theranostics as an alternative treatment for patients with recurrent grade 3 and grade 4 gliomas, with an estimated completion date in December 2025 (NCT05644080).
bioengineering-11-00714-t003_Table 3Table 3Clinical trials investigating PSMA-RLT in non-prostatic cancers (ClinicalTrials.gov).Type of CancerCentre/SponsorPatientsStudy PhaseTrial ID(Reference)StatusAdenoid cystic carcinomaPeking Union Medical College Hospital, Beijing, China40Early Phase INCT04801264UnknownRenal cell carcinomaPeking Union Medical College Hospital, Beijing, China40Not ApplicableNCT05170555UnknownSalivary gland cancerRadboud University Medical Center, Nijmegen, Gelderland, Netherlands12Phase IINCT04291300CompletedPSMA-positive tumoursIstituto Scientifico Romagnolo per lo Studio e la Cura dei Tumori, Cesena, Italy100Phase IINCT05867615RecruitingHigh-grade gliomaSt. Olavs Hospital, Trondheim, Norway10Not ApplicableNCT05644080RecruitingSoft tissue sarcomaUniversity of Lausanne Hospitals, Lausanne, Vaud, Switzerland20Phase INCT05420727RecruitingMetastatic clear cell renal cancerCentre Leon Berard, Lyon, France48Phase I/IINCT06059014Recruiting


## 4. Discussion and Conclusions

The process of angiogenesis plays an important role in tumour growth and the metastatic spread of cancers [45]. Selective inhibition of tumour growth by anti-angiogenic drugs represents a potential treatment strategy and is considered an open field of research [46]. In this scenario, the expression of PSMA by the endothelium of the newly formed vasculature could also be used as a target in PSMA-avid non-prostate tumours [47,48]. In addition, it has been demonstrated that PSMA is expressed not only in endothelial cells but also in different cell types [27,49]. Under these premises, its feasibility has been investigated in preclinical and clinical studies, especially in some cases of poor prognosis and limited therapeutic options.

Despite the very preliminary results, PSMA could be considered an interesting target in triple-negative breast cancer cells. PSMA-RLT has been shown to significantly reduce cell viability and induce an anti-angiogenic effect, especially under hypoxic conditions that increase PSMA binding and uptake [27,28,29]. However, the poor preclinical evidence of PSMA-RLT efficacy narrows the development of PSMA-targeted imaging and therapeutic approaches for other tumour models.

Clinical studies with a small sample size have shown heterogeneous results. In salivary gland malignancies, PSMA-RLT achieved disease stabilization without major adverse events in only a few patients. However, the frequent discontinuation due to unsatisfactory tumour-absorbed doses and the shorter PSMA retention time than in PCa cells indicate that PSMA-RLT for salivary gland tumours needs further improvement [31]. As a palliative treatment after failure of other palliative options, PSMA-RLT could be considered for recurrent and/or metastatic salivary gland malignancies to achieve temporary disease stabilization, relief of tumour-related discomfort, and improved quality of life [32]. It has also been reported that in metastatic adenoid cystic carcinoma, [^177^Lu]Lu-EB-PSMA-617 may lead to an improvement in both clinical symptoms and imaging response, as well as significant favourable outcomes in recurrent foci and liver and lung metastases, while doubts remain on the possible efficacy of PSMA-RLT on bone metastases.

In HGG, the efficacy of PSMA-RLT remains questionable because the tumour dose achieved is too low for a sufficient therapeutic effect compared with external beam radiotherapy [35]. In the study by Truckenmueller et al., PSMA-RLT was considered as last-line therapy after recurrence in HGG patients selected based on [^68^Ga]Ga-PSMA PET/MRI, and only a small proportion of patients were considered eligible. A lower TBRmax threshold and earlier stage of disease are suggested to be considered as inclusion criteria in future studies [36]. In addition, a cocktail of ^177^Lu and 225Ac-PSMA has been proposed in HGG patients, combining the wider range and crossfire effects of beta emitters with the higher energy levels and fewer side effects of alpha emitters [35]. Similarly, PSMA-RLT was not effective in HCC patients with no other treatment options because of an inadequate tumour radiation dose [37].

Although heterogeneous, the results observed in patients with thyroid cancer are more encouraging. PSMA-RLT showed a mild and transient response without serious side effects in a few selected metastatic thyroid cancer patients, including radioactive iodine-refractory DTC, ATC, and PDTC, and could be considered a therapeutic option in case of progression when established therapies are no longer effective, considering the lack of promising alternatives in these cases [38,39]. High PSMA expression was detected in dedifferentiated thyroid carcinoma with more aggressive behaviour, which is related to tumour autoregulation that facilitates endothelial cell invasion, angiogenesis, and neovascularization. The preliminary studies showed intense radiopharmaceutical accumulation near tumour cells in early and late scans after radioligand injection, suggesting that the neovasculature of ATC and PDTC microenvironment represents a sufficient target because of a long retention time [38,39]. The less-than-encouraging results in most of the solid non-prostate malignancies studied are related to the low or absent PSMA expression in tumour cells, while it is predominant in the endothelium of the tumour-associated neovasculature [36]. The localization of PSMA expression in the neovasculature of salivary gland malignancies, high-grade glioma, hepatocellular carcinoma, and thyroid carcinoma, rather than in the cytoplasm and membrane of PCa cells, may explain why PSMA-RLT does not achieve sufficient radiation distribution within the tumour to be therapeutically targetable and effective compared with PCa [34,40]. In addition, the strong heterogeneity in PSMA expression between primary and metastatic sites (along with the different amounts of neovasculature) needs to be considered [36,41,50].

Further research should reveal whether PSMA-avid patients without prostate cancer could benefit from a higher activity regimen or a shortened interval of PSMA-RLT [32,35]. In addition, stricter patient selection, such as focusing on specific histopathologic subtypes, may improve the PSMA-RLT response [32].

Some authors suggested that the introduction of RLT with alpha-emitting agents such as [^225^Ac]Ac-PSMA-617 may be more successful because of the higher linear energy transfer [31,32], thus minimizing systemic toxic effects [35,51,52]. However, the shorter path length of alpha-emitters may compromise the tumoricidal radiation effect in the case of endothelial PSMA expression, in the non-prostate tumour, since the main energy deposition would occur in the neovascular cells rather than the tumour cells, resulting in hypoxia and radioresistance. Graef et al. suggested that a cocktail of [^177^Lu]Lu-PSMA and [^225^Ac]Ac-PSMA could be advantageous in this case [35].

Ongoing clinical trials are investigating PSMA-targeted therapeutic approaches in other PSMA-avid non-prostate cancers. Strong PSMA expression and high tracer accumulation have led to the consideration of PSMA-RLT as an interesting therapeutic option that could benefit metastatic clear cell RCC patients.

The need for larger patient cohorts and more RLT cycles administered underscores the need for further comprehensive studies. Given the very preliminary results of both preclinical and clinical studies, clinical trials in the near future may provide stronger evidence regarding both the safety and therapeutic efficacy of PSMA-RLT in malignancies other than prostate cancer.

## Figures and Tables

**Figure 1 bioengineering-11-00714-f001:**
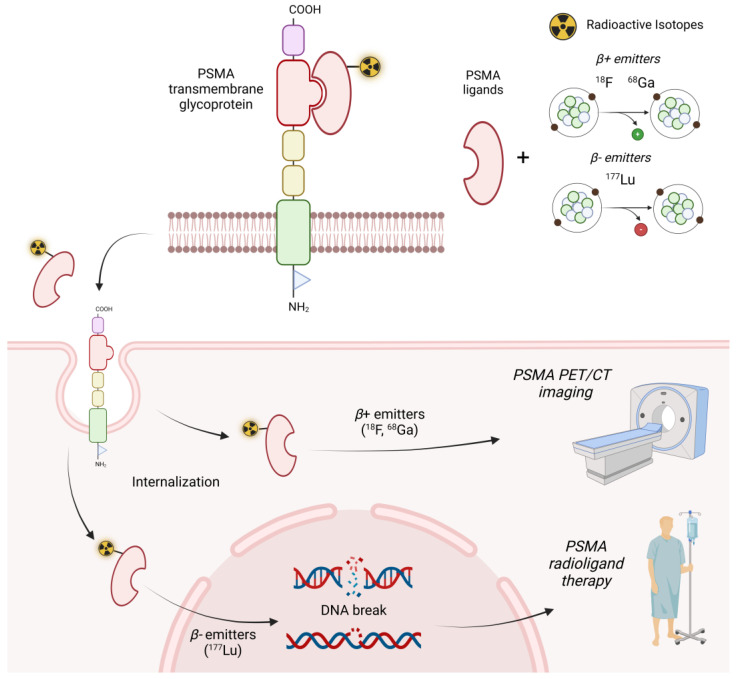
Schematic representation of the theragnostic approach with prostate-specific membrane antigen (PSMA). The figure was created with BioRender.com.

## Data Availability

Not applicable.

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
