# Peer review of "Prostate-Specific Membrane Antigen Radioligand Therapy in Non-Prostate Cancers: Where Do We Stand?"

_bioengineering, 2024, doi:10.3390/bioengineering11070714_

Round 1

Reviewer 1 Report

Comments and Suggestions for Authors

Anna Giulia Nappi et al. have submitted a concise and timely review on PSMA-RLT of non-prostate cancers, which covers preclinical and clinical studies and ongoing clinical trials. The review is well written and well organized, being certainly interesting and comprehensible for a wide range of readers that do not belong to the Nuclear Medicine field. I am glad to recommend the publication of the manuscript in “bioenginnering”. However, before publication, the authors should address the following issues:

1) Page 2, line 85: I suggest to include also the commercial name of [177Lu]-PSMA-617, i.e. PluvictoTM.

2) Page 3, Figure 1: The figure depicts the general structure of a radiolabeled antibody to illustrate PSMA-targeted radioprobes. This is not very adequate as the review focuses only in PSMA antagonists of the Lys-urea-Glu type, which correspond to small molecules. This can lead to misleading interpretations and need to be changed by introducing a generic representation of the labeled antagonist instead of the antibody.

3) Page 4, line 127: Please, correct the typo “luthetium-117”.

4) Page 5, line 208: Please, define AURA.

5) Page 7, Line 298: Replace 6000 MBq by 6.0 GBq to uniformize with the other dose values provided in the review that are all expressed in GBq.

6) Page 5, Caption of Table 1: I suggest to remove the word “characteristics” from the caption.

7) Page 8, Caption of Table 2: I suggest to remove the word “characteristics” from the caption.

8) Page 8, Table 2: In the entry “Type of disease” uniformize the use of capital letters.

9) Page 9, Table 3: Please, add the country to each clinical trial.

Author Response

ANSWERS TO REVIEWER #1

We would like to express our sincere thanks to the Reviewer#1 for the constructive suggestions and positive comments.

Here, we provide our point-by-point answer:

Reviewer #1: 1. Page 2, line 85: I suggest to include also the commercial name of [177Lu]-PSMA-617, i.e. PluvictoTM.

Action. Done (line 84 page 2).

Reviewer #1: 2. Page 3, Figure 1: The figure depicts the general structure of a radiolabeled antibody to illustrate PSMA-targeted radioprobes. This is not very adequate as the review focuses only in PSMA antagonists of the Lys-urea-Glu type, which correspond to small molecules. This can lead to misleading interpretations and need to be changed by introducing a generic representation of the labeled antagonist instead of the antibody.

 Action. Thank you for your suggestions. The figure 1 in page 3 was changed and reupload according to your suggestion, introducing a generic representation of the labeled antagonist instead of the antibody.

Reviewer #1: 3. Page 4, line 127: Please, correct the typo “luthetium-117”.

Action. Thank you for your suggestions. However, I can't find this typo in the file. it was probably already changed in a revised version.

Reviewer #1: 4. Page 5, line 208: Please, define AURA.

Action. As suggested, I introduce the definition of both AURA and mAURA in Page 5 lines 212-213.

Reviewer #1: 5. Page 7, Line 298: Replace 6000 MBq by 6.0 GBq to uniformize with the other dose values provided in the review that are all expressed in GBq.

Action. Done (line 308 page 7).

Reviewer #1: 6. Page 5, Caption of Table 1: I suggest to remove the word “characteristics” from the caption.

Action. Done.

Reviewer #1: 7. Page 8, Caption of Table 2: I suggest to remove the word “characteristics” from the caption.

Action. Done.

Reviewer #1: 8. Page 8, Table 2: In the entry “Type of disease” uniformize the use of capital letters.

Action. Done.

Reviewer #1: 9. Page 9, Table 3: Please, add the country to each clinical trial.

Action. Thank you for your suggestions. I added the country to each clinical trial In Table 3 Page 9.

Reviewer 2 Report

Comments and Suggestions for Authors

Dear authors,

thank you for your interesting work about non prostate PSMA use for NM diagnostics and therapy. It is a very hot topics as shown by the last JNM content.

A very important point is that the text you wrote lets to think that only endothelial cells of non-prostate tumors are able to express superficial PSMA moities, that seems no really true (J  Nucl Med 2024 65:1004-1006 (10.2967/jnumed.123.266659)). So please modulate your text in this direction.

In an other point of view, have you an idea of the biological mechanism of PSMA expression on endothelial surface of some cancers? Is it a specific way or a general property of endothelial cancer cells, even if too weakly expressed to be detectable in many cancer types ?

Furthermore, line 415, the fact that endothelial cells would be irradiated by alpha emitters is not a strong argument because the main exnergy deposition would be in these endothelial cells (with subsequent hypoxia and radioresistance) and not really in tumor cells. This scepticism does not into account immunological activation consequences, a domain not aborded in this work.

Line 147, you mention tumour associated endothelial cells and proposed HUVEC for abbeviation. I think it would be better to reserve this acronym to umbilical vein endothelial cells as mentionned line 153.

Sincerely yours

Author Response

ANSWERS TO REVIEWER #2

We would like to express our sincere thanks to the Reviewer#2 for the constructive suggestions and positive comments.

Here, we provide our point-by-point answer:

Reviewer #2: 1. A very important point is that the text you wrote lets to think that only endothelial cells of non-prostate tumors are able to express superficial PSMA moities, that seems no really true (J  Nucl Med 2024 65:1004-1006 (10.2967/jnumed.123.266659). So please modulate your text in this direction.

Action. The discussion has been modulated in the direction that you suggested and the article you proposed was added as a reference.

Reviewer #2: 2. In an other point of view, have you an idea of the biological mechanism of PSMA expression on endothelial surface of some cancers? Is it a specific way or a general property of endothelial cancer cells, even if too weakly expressed to be detectable in many cancer types?

Action. This is a really interesting comment and for sure a central point of the review. Currently, we think that the data available in literature are still not strong enough to provide a clear and definitive answer to this question, since different factors needs to be taken into account and an answer could reveal only an idea not based on clear evidences. Surely, a wide variety of immuno-histochemistry studies have shown PSMA to be upregulated on the endothelial cells of the neovasculature of a wide variety of different solid tumors where it may facilitate endothelial cell sprouting and invasion through its regulation of lytic proteases that have the ability to cleave the extracellular matrix. However, as mentioned in the paper, the poor preclinical evidence of PSMA-RLT efficacy narrows the development of PSMA-targeted imaging and therapeutic approaches for other tumor models.

Reviewer #2: 3. Line 415, the fact that endothelial cells would be irradiated by alpha emitters is not a strong argument because the main energy deposition would be in these endothelial cells (with subsequent hypoxia and radioresistance) and not really in tumor cells. This scepticism does not into account immunological activation consequences, a domain not aborded in this work.

Action. Thank you for your observation. In page 10, lines 432-439 were improved as suggested. The concept of less therapeutic efficacy of alpha-emitter agents in case of endothelial PSMA expression in non-prostate tumor has been clarified, as suggested by Graef et al study included in our review (doi: 10.2967/jnumed.122.264850). In the discussion, we comment and discuss only the results of the included studies without speculating on further hypotheses, for example relating to immunological activation.

Reviewer #2: 4. Line 147, you mention tumour associated endothelial cells and proposed HUVEC for abbeviation. I think it would be better to reserve this acronym to umbilical vein endothelial cells as mentionned line 153.

 Action. Thanks to the Reviewer’s observation. I removed the acronym HUVEC from page 4, line 149, and specified in lines 155-156 that the researcher examined PSMA expression on the endothelial cell line isolated from human umbilical vein, which has the acronym HUV-EC- C, but in the study the authors referred to this cell line as HUVEC and we also used this acronym in this manuscript.

Reviewer 3 Report

Comments and Suggestions for Authors

This is an excellent review article covering the use targeting PSMA for non-prostate cancer imaging and treatment. There are few preclinical and clinical investigations looking at PSMA outside of prostate cancer, so this is a timely article.

This article is very well written and the authors do an excellent job of describing each study referenced with an appropriate amount of detail. 

Suggestion: clarification of the difference between targeting the vasculature or cell lines being PSMA overexpressing. Referring to lines 150-142 on page 4. This reviewer was confused as to how TNBC cell lines could be targeted with PSMA in vitro, yet, the authors say that only the vasculature can be targeted and that there is a lack of these models. This is not to say that the authors are incorrect, but it certainly was unclear to this reviewer what is being targeted in the TNBC cells, if not the cells themselves.

Minor grammatical errors and typos:

1) P. 4, line 135 the word table should be capitalized and plural "Tables 1-3"

2) P. 4, line 161, 177Lu should have 177 as superscript.

3) P. 5, line 218, 177Lu should have 177 as superscript.

4) P. 10, line 373, 177Lu should have 177 as superscript

             line 380, 68Ga should be 68 as superscript

             line 383, 177Lu should have 177 as superscript.

Please review the manuscript carefully to ensure this formatting is correct throughout.

Author Response

ANSWERS TO REVIEWER #3

We would like to express our sincere thanks to the Reviewer#3 for the constructive suggestions and positive comments.

Here, we provide our point-by-point answer:

Reviewer #3: 1. Clarification of the difference between targeting the vasculature or cell lines being PSMA overexpressing. Referring to lines 150-142 on page 4. This reviewer was confused as to how TNBC cell lines could be targeted with PSMA in vitro, yet, the authors say that only the vasculature can be targeted and that there is a lack of these models. This is not to say that the authors are incorrect, but it certainly was unclear to this reviewer what is being targeted in the TNBC cells, if not the cells themselves. 

Action. First, we would like to underline that the point of the expression of PSMA not only by endothelial cells has been addressed by responding to the question of another Reviewer, therefore some parts of the discussion have been changed according to this direction. We made little changes to the lines that you mentioned in order to make them clearer.

Reviewer #3: 2. P. 4, line 135 the word table should be capitalized and plural "Tables 1-3".

Action. Changed.

Reviewer #3: 3. P. 4, line 161, 177Lu should have 177 as superscript. 

Action. Changed.

Reviewer #3: 4. P. 5, line 218, 177Lu should have 177 as superscript.

 Action. Changed.

 Reviewer #3: 5. P. 10, line 373, 177Lu should have 177 as superscript; line 380, 68Ga should be 68 as superscript; and line 383, 177Lu should have 177 as superscript.

Action. Changed.
